# Mechanical Strength and Surface Analysis of a Composite Made from Recycled Carbon Fibre Obtained via the Pyrolysis Process for Reuse in the Manufacture of New Composites

**DOI:** 10.3390/ma17020423

**Published:** 2024-01-14

**Authors:** Rita C. M. Sales-Contini, Hugo M. S. Costa, Heide H. Bernardi, William M. M. Menezes, Francisco J. G. Silva

**Affiliations:** 1Technological College of São José dos Campos-Professor Jessen Vidal, Centro Paula Souza, Avenida Cesare Mansueto Giulio Lattes, 1350 Distrito Eugênio de Melo, São José dos Campos 12247-014, SP, Brazil; heide.bernardi@fatec.sp.gov.br (H.H.B.); william.menezes@fatec.sp.gov.br (W.M.M.M.); 2Department of Aeronautics, Laboratory of New Concepts in Aeronautics, Aeronautics Institute of Technology, Praça Marechal Eduardo Gomes 50, Vila das Acácias, São José dos Campos 12228-900, SP, Brazil; hugomoreiracosta@hotmail.com; 3Instituto Superior de Engenharia do Porto, Polytechnic of Porto, Rua Dr. António Bernardino de Almeida, 4249-015 Porto, Portugal; fgs@isep.ipp.pt

**Keywords:** carbon fibre, recycling, pyrolysis, mechanics properties, product development, sand ladder platform

## Abstract

This work aims to obtain recycled carbon fibre and develop an application for this new material. The carbon fibres were obtained by recycling aerospace prepreg waste via the pyrolysis process. The recycled fibres were combined with an Araldite LH5052/Aradur LY5053 epoxy resin/hardener system using manual lay-up and vacuum bagging processes. For comparison, the same resin/hardener system was used to produce a composite using commercial carbon fibre. The recycled and commercial composites were subjected to flexural, tensile and Mode I testing. Fracture aspects were analysed via scanning electron microscopy (SEM). The pyrolysis process did not affect the fibre surface as no degradation was observed. The fracture aspect showed a mixture of failure in the recycled composite laminate and interlaminar/translaminar failure near the surface of the commercial composite caused by flexural stress. Flexural and tensile tests showed a loss of mechanical strength due to the recycling process, but the tensile values were twice as high. The sand ladder platform was the project chosen for the development of a product made with recycled carbon fibres. The product was manufactured using the same manufacturing process as the specimens and tested with a 1243 kg car. The method chosen to design, manufacture and test the prototype sand ladder platform made of recycled carbon fibre was appropriate and gave satisfactory results in terms of high mechanical strength to bending and ease of use.

## 1. Introduction

In some industries, such as aerospace, automotive and sports, there has been an increase in the use of composite materials to produce more efficient structures and more complex designs [1]. Carbon fibre (CF) is an example of a high-performance material reinforcement, which is the most widespread reinforcement used in advanced composites [2]. The carbon fibre-reinforced composite is a new engineering material, which is growing in popularity due to its high strength and low density. Although carbon fibres have been expensive for many applications in the past, the manufacturing cost has dropped significantly in the last decade, while the production volume and global demand have increased. In this way, carbon fibres have become the most used in the aerospace industry and spread to other areas of industry [3]. 

The composites used in the aerospace industry are generally prepregs with a thermally stable polymer matrix (epoxy, unsaturated polyester or phenolic resin) reinforced with glass or carbon fibre [4], with the carbon fibre-reinforced composite being the most used. As an example, the Boeing 787 and Airbus A350 gain more than 50% of their weight from this type of material [5].

Nowadays, the aerospace industry uses large-scale carbon fibre-reinforced plastics (CFRP), but the increase in the use of this type of material brings an increase in its residues, generating environmental and economic concerns related to its residual disposal [6]. These residues can be generated via the loss of the prepreg shelf life, manufacturing scraps and specimens of the tested materials [7]. 

The recycling of materials is a factor that has generated a lot of discussion among nations because this action can reduce the exploitation of the environment to obtain resources and reduce the contamination of water and soil with toxic waste generated by discarded materials. A more viable sustainable solution would be to recycle the waste generated by the process itself, which would generate savings for the industry and create a new production process. The 2030 Agenda has 17 key goals for nations to consciously achieve sustainable development to better preserve the planet by 2030. Material recycling is in line with Goal 12 of the 2030 Agenda, which implores that nations, businesses and institutions “ensure sustainable consumption and production patterns” [8].

It is estimated that the volume wasted in Europe in 2009 was 22,750 tons, and in the United States, an average of 13,500 tons of carbon fibre is disposed of in landfills annually. In Europe, this number reaches 23,100 tons. The recovery of these materials would eliminate the sending of scrap materials to landfill and the incineration of waste [9,10].

The best way to solve this problem that leads to better environmental and economic benefits is via the recycling process [11,12]. As carbon fibres are a high-technology material that needs an expensive manufacturing process (which is reflected in the fibre cost), the process of recycling residues of carbon fibre-reinforced composites makes sense if carried out in an effective and economically viable way [13]. 

According to Witik et al. [9], in the European Union, there are already robust policies that help to reduce impacts of these materials on the environment, establishing four main stages for the treatment of waste: prevention, reuse, recycling and the restriction of the use of landfill and incineration. In 2015, legislation increased the percentage of minimum recycling allowed for each vehicle to 85%, that is, all this percentage of the material must be recycled and reused as secondary raw material. Therefore, development from an ecological point of view makes perfect sense going forward, taking into account the maintenance and possibility of reusing resources.

To overcome this challenge, several companies met in groups to develop methods to recycle this type of polymeric material, such as ERCOM Composite Recycling GmbH (Germany), VALOR (France), SMC Alliance (USA) and the FRP Forum (Japan) [14] (GOODSHIP, 2007). In this scenario, solid waste management and good practices prove to be highly profitable for companies since their techniques of reduction at source, substitution of raw materials and reuse can bring real economic benefits, in addition to avoiding business exposure to liabilities and environmental risks (the devaluation or loss of activity). The waste related to carbon fibres will quickly reach a level of significance for the environmental issue because it is not biodegradable. Also, give the concern for the environment, both in terms of limiting the use of finite resources and the need to manage waste disposal, it has led to increased pressure to recycle materials at the end of their useful lives. Recycling operations are already being established and driven by the economy [15].

Recycled carbon fibre has the potential to be cheaper than commercial fibre, and this has opened up a new market and new opportunities in different industries. This is already being recognised and has led to the development for several fields of application such as electrical conductors for electromagnetic shielding, fabrics for thermal clothing and reinforcements for ceramic brake discs [16].

Boeing, in a project with the Aircraft Fleet Recycling Association (AFRA), presented the viability of the reused fibres by developing an “armrest” made only from recycled CF derived from the pre-production residues of the fuselage of the Boeing 787 Dreamliner. In the international scenario, recycling gained other proportions, pushing the two biggest competitors, Airbus and Boeing, in 2008, to optimise the process of the recycling and recovery of aircraft materials, consequently reducing the amount of waste discarded at the end of the product cycle [17].

Other applications of recycled carbon fibre composites are also being explored. Meredith [18] applied recycled carbon fibre-reinforced composite fabric (rCFRP) to non-critical parts of an environmentally sustainable Formula-3 car. 

Another sector where recycled composites can be used is defence. The defence sector requires the development of tools and accessories made of lightweight materials that meet mechanical, thermal, fire retardant and infiltration requirements to carry out search and rescue missions in the event of natural disasters or those caused by war. In these cases, it is necessary to transport items, such as reinforced boxes containing medicines, first aid kits, stretchers and even sand ladder platforms. The sand ladder platform product is used in rescue situations where the vehicle is travelling over unstable terrain, such as soft or slippery ground. The product is placed under the wheels on the slippery terrain to provide the necessary traction to move the vehicle away from the obstacle. The most common platforms are aluminium military platforms, which have been the standard method for many years, but their supply has been limited. As a result, various companies in the UK and Germany began to produce their own versions, using a lattice-like design made from materials such as fibreglass and Kevlar [19].

To recover these materials, processes such as mechanical recycling [20,21], chemical recycling [1], electrochemical recycling [22] and energy recycling [6,7], or a combination of these methods, have been used to recover carbon fibres and transform them into new materials. However, the most widespread method for recycling CFRP that is proven to be realistic, practical and cheaper is known as pyrolysis [23,24]. This method consists of subjecting the material to high temperatures (300–800 °C) in an inert atmosphere for the degradation of the polymer matrix and generation of recovered fibres [12]. Nevertheless, after the pyrolysis process, the recovered fibres do not reliably present the properties of commercial fibres, and the composites generated from recovered carbon fibre present a loss of mechanical strength of 10–15% compared to composites produced with commercial raw materials [3]. 

Therefore, to meet the need for waste control, this work aims to develop a sand ladder platform product from recycled carbon fibre. Before product development, the polymer composite specimens were prepared using recycled scraps (obtained via pyrolysis) of prepreg waste from the aerospace industry impregnated with epoxy resin. The recycled carbon fibre-reinforced polymer composite (rCFRP) samples were characterised. Mechanical tests, density analysis and morphological surface analysis via SEM were carried out. The results of the (rCFRP) samples were compared with those of commercial carbon fibre-reinforced polymer composites (cCFRP). Finally, to validate the feasibility of using carbon fibre for new product development, the sand ladder platform was fabricated and tested.

## 2. Materials and Methods

### 2.1. Pyrolysis

In the pyrolysis process, the pre-impregnated residues (Hexcel 8552/AS4) were thermally treated at 500 ± 10 °C for 4 h in an argon atmosphere using an Inforgel furnace. Figure 1 shows the pre-impregnated residue images as received and placed into the oven before the pyrolysis process. 

### 2.2. Specimen Preparation

For the mechanical tests, the first set of test specimens was prepared from laminate manufactured with the plain wave fabric, AS4 3K 200 g/m^2^ (HexPly^TM^, Salt Lake City, UT, USA) [25] recycled via the pyrolysis method, combined with resin/hardener system Araldite LY 5052 (Huntsman^TM^, Basel, Switzerland) and Aradur 5052 (Huntsman^TM^, Basel, Switzerland) in a mix ratio of 100:38 parts by weight. The second set of test specimens was prepared using a commercial carbon fibre AS4 3K 200 g/m^2^ (Texiglass^TM^, Vinhedo, SP, Brazil), and the resin/hardener system was the same as that previously used.

The fabricated laminates were named as follows: (a) rCFRP—recycled carbon fibre-reinforced polymer composite; (b) cCFRP—commercial carbon fibre-reinforced polymer composite. The laminates were manufactured via the hand lay-up process (HLUP), followed by the vacuum bagging application (VB). 

Table 1 presents the laminate plate dimensions and laminate stack sequence of each composite. All the laminates were cured at 60 °C for 2 h and post-cured at 80 °C for 2 h in a Makel double electric oven according to the resin data sheet [26]. After completing the curing process, the cured laminates were demoulded and cut into specimens to conduct the flexural and tensile tests according to the ASTM standards (Table 1).

### 2.3. Density and Porosity Measurements

For the density measurement, both rCFRP and cCFRP test specimens were weighed by using an analytical balance Sartorius. The tests were performed according to the Archimedes method following Procedure A of ASTM D792-08 [29]. To measure the composite density, the specimen mass was determined in the air, and it was immersed in a liquid to determine its apparent mass upon immersion, and then its specific gravity was calculated. The test specimen was 1 g–50 g, and the temperature during the test was 23 ± 2 °C. For comparison, the composite density was also measured through the helium pycnometer method using an Accupyc II 1340 instrument (Micromeritics, Norcross, GA, USA).

From the density values obtained using the previous methods, it was possible to perform calculations of the fibre volume fraction and the void content following the ASTM D3171-11 [30]. We also estimated the product mass and expected mechanical properties using the law of mixtures. Equation (1) presents the calculation of the estimated mass of the product, where m_p_ is the estimated mass of the product, V_p_ is the volume of the product obtained using the model designed in the Solid Edge 2019 software, and d_cr_ is the density of the rCFRP. Then, Equation (2) shows the calculation of the fibre volume fraction (v_f_) according to the resin density and CF data. Here, d_r_ is the density of the resin, d_F_ is the densities of the cCFRP and rCFRP and d_C_ is the density of the composite with cCFRP or the rCFRP.

Equation (3) was used to calculate the modulus of elasticity estimated for the composite from the mixtures law, where E_e_ is the estimated modulus of elasticity and E_f_ is the modulus of elasticity of the fibre; this value was taken for a commercial fibre from the technical data provided by the manufacturer (AS4 3K 200 g/m^2^/HexPly^TM^, Salt Lake City, UT, USA), and for recycled fibre, the same value was multiplied by the loss in modulus obtained from the flexural test results). v_f_ is the volume fraction of the fibre and E_m_ is the modulus of elasticity of the matrix (Epoxy resin 8552, HexPly^TM^, Salt Lake City, UT, USA), according to the technical data provided by the manufacturer.
(1)$$m_{p}=V_{p}\cdot d_{cr}$$
(2)$$d_{c}=\left(d_{f}\cdot v_{f}\right)+(d_{r}\left(1-v_{f}\right))$$
(3)$$E_{e}=\left(E_{f}\cdot v_{f}\right)+{(E}_{m}\left({1-v}_{f}\right))$$

### 2.4. Mechanical Tests

#### 2.4.1. Flexural Test

The specimens were tested according to ASTM D790-10 [27]. This test procedure consists of positioning a bar of the rectangular cross-section under two supports with a delimited distance and applying a load using a load nose positioned midway between the supports. To perform the flexural test, an Instron^TM^ test machine (Norwood, MA, USA) with a load capacity of 100 kN at a test speed of 2 mm/min was used. The test was performed at room temperature and relative humidity of 50% and only finished when it reached a maximum strain of 5.0% or a rupture occurred on the outer surface of the test specimen.

#### 2.4.2. Tensile Test

The specimens were tested by following ASTM D3039/D3039M-08 [28]. This test method determines the in-plane tensile properties of polymer matrix composite materials reinforced with high-modulus fibres. In this test, a thin, flat strip of material with a constant rectangular cross-section was mounted in the grips of a mechanical testing machine and monotonically loaded in tension while recording the force. The test was performed using an Instron^TM^ test machine with a load capacity of 100 kN at a test speed of 2 mm/min.

#### 2.4.3. Mode I Fracture Toughness Characterization

The crack propagation of the rCFRP specimen was measured using a high-definition camera with a macro lens attached to it. The samples were white-painted on both sides to enhance the crack tip visualisation. Vertical lines were drawn every 1 mm from the end of the Teflon^®^ insert, where the delamination process should start, until 50 mm and then after every 5 mm for the following 30 mm.

For all tests, aluminium loading blocks were used to apply the load to the samples. Each block was bonded with epoxy adhesive in the specimens’ ends, where the TeflonTM insert was placed. 

The samples were then attached to an InstronTM 5500R testing machine (Norwood, MA, USA) with a loading cell of 2 kN. The load was applied with a constant displacement rate of 1 mm/min. The tests were conducted at 25 °C and a relative humidity of 50%. The delamination extension as a function of the applied load was registered using a crack marker, which was pressed every time the crack crossed one of the vertical marks made in the lateral of the samples, which was observed using an ImetrumTM Video Gauge 3.0 camera placed in front of the test. The energy release rate expression, as given in Equation (4), was obtained from the beam theory for a double cantilever beam according to ASTMD5528-01 [31], considering the correction factors for both rotations at the delamination front and large displacement effects. The equation is expressed as follows:(4)$$G_{I}=\frac{3P\delta F_{c}}{2w(a+\left|\Delta \right|)}$$
where P is the applied load, δ is the transversal displacement where the load is applied, α is the crack length, w is the sample width, Δ is the correction for the rotation that may occur at the delamination front and F_c_ is the correction factor for large displacements [31].

### 2.5. Scanning Electron Microscopy Analysis

The pyrolysis process efficiency was evaluated using the observation of the recycled fibre surfaces via high-resolution scanning electron microscopy with the field emission gun (FEG) MIRA3 TESCAN (Kohoutovice, Czech Republic). The analyses were performed with an acceleration energy of 10–30 keV, working distance of 19–28 mm, beam intensity of 7–18 W/cm^2^ and magnification range of 4–40 kX, and all the images were obtained using a secondary electron detector.

For the mechanical test specimens’ fracture analyses, the specimens were covered with a thin gold layer via vacuum evaporation using QUORUM–Q150RE5 equipment (East Sussex, UK). The SEM analyses were performed using VEGA3 XMU TESCAN equipment (Kohoutovice, Czech Republic) with a secondary electron detector, an acceleration energy of 5 keV, a working distance of 9–10 mm and a magnification range of 50–500×.

### 2.6. Sand Ladder Platform Project

#### 2.6.1. Sand Ladder Platform Dimensioning

As a premise for executing the sand ladder platform project, it was considered that the effort employed in its use would only be the bending effort, which is why the data generated via the bending test were used to design it. Furthermore, it was considered that the load suffered by the platform (P_Platform_) would be static and equal to the weight of an EcoSport car (Model used for field testing) divided by four, considering that the weight of the car is divided equally between the four wheels, as illustrated in Equation (5), where P_car_ is the weight of the EcoSport model car. Table 2 presents the weight value of a Ford EcoSport Freestyle 1.6, according to the technical data sheet provided by Ford [32].
(5)$$P_{Platform}=\frac{P_{Car}}{4}$$

Based on these premises, Equation (6) was used to calculate the maximum stress (σ_max_) suffered by the sand ladder platform according to pre-defined width and length dimensions based on analysis of similar products on the market, and in Equation (6), M_max_ is the maximum moment, y is the neutral axis of the cross-section and I is the moment of inertia.

The maximum stress (σ_max_) obtained using Equation (6) was related to the breaking stress (σ_rc_) generated via the flexion test using Equation (7) to establish the safety factor (FS). The FS value was considered ideal if it was greater than 3.14, following the concept in Equation (8). In Equation (8), FS_material_ is the FS related to the contribution of the material, FS_loading_ is the FS related to the contribution of loading, FS_geometry_ is the FS related to the geometry contribution, FS_failure analysis_ is the FS related to the failure analysis contribution and FS_reliability_ is the FS related to the reliability contribution; this value was chosen to generate a dimension, taking into account the reliability of the product and the extreme situations that it was subject to when in use [33].

To check whether the maximum stress estimated from Equation (6) would correspond to a maximum deformation consistent with the properties obtained for the composites with recycled CF from the flexural test, the calculation of the maximum deformation suffered by the sand ladder platform was carried out using Equation (9). In Equation (9), $\varepsilon_{\max}$ is the maximum deformation suffered by the platform, and E_BR_ is the average value of the tangent modulus of elasticity for composites produced with recycled CF [33].
(6)$$\sigma_{max}=\frac{M_{max}\cdot y}{I}$$
(7)$$FS=\frac{\sigma_{rc}}{\sigma_{max}}$$
(8)$$FS={FS}_{material}\cdot {FS}_{loading}\cdot {FS}_{geometry}\cdot {FS}_{failureanalysis}\cdot {FS}_{reliability}$$
(9)$$\varepsilon_{max}=\frac{\sigma_{max}}{E_{BR}}$$

#### 2.6.2. Mould Design

To manufacture the lamination mould for the product “Sand Ladder Platform”, additive manufacturing was chosen, using Sethi3D AiP A3- 1.75 mm equipment (Campinas, SP, Brazil), installed at Technological College of São José dos Campos (São José dos Campos, SP, Brazil), and ABS MG94 (Essentium, Pflugerville, TX, USA) as the material. It had a bluish-white colour. Mould design is made up of four sections of equal dimensions joined together using 6 × 30 mm wooden dowels glued with professional araldite with a fill factor of 10% and an internal structure made up of rectangular honeycombs. Furthermore, the mould was equipped with a structure for applying a vacuum bag more efficiently for the application of intermediate vacuums used for the better compaction of the layers, which were also manufactured using the same technique and composed of 5 sections. 

#### 2.6.3. Manufacturing of the Product “Sand Ladder Platform”

The manufacturing of the “Sand Ladder Platform” product was carried out at the Technological College of São José dos Campos (São José dos Campos, SP, Brazil) and followed the lamination procedure used to manufacture the flexural test specimens. The distinction is the application of intermediate vacuums for every 10 laminated layers. Using the mould, 5 equal sections were manufactured and joined together using the same material (recycled carbon fibre and epoxy resin). The thickest part consisted of 56 sheets of recycled carbon fibre stacked at 90° with epoxy resin (Araldite^®^ LY 5052/Aradur^TM^ 5052, Huntsman^TM^, Basel, Switzerland). After laminating each section and joining them, the resin curing cycle was carried out, and the piece was demoulded for finishing. 

## 3. Results

### 3.1. Microscopy Analysis

Figure 2 shows SEM images of prepreg scrap as received and the fibres after the pyrolysis process to compare the effects of the pyrolysis process. In the prepreg scrap, the cured resin can be seen on the surface of the carbon fibre fabric (Figure 2a), and after the pyrolysis process, some particles of resin degradation, which look like dirt, can be seen on the surface of the fabric (Figure 2b).

Figure 3 shows the SEM images of the pyrolysed fibre surface used to analyse possible damages caused by the recycling process. The SEM images showed that the pyrolysis process was efficient. The surface of the fibres has no damaged areas, and the visible stretch marks on the surface of the carbon fibre, as highlighted by arrows in Figure 3a, can be related to the manufacturing process of the fibres; these are created during the extrusion stage.

In Figure 3b, it is possible to observe the presence of the sizing region, as highlighted by circles and identified by white regions inside the stretching marks and residues of the polymeric matrix (highlighted by arrows) on the fibre surface. The sizing is a superficial treatment applied on the fibres, which generally consists of polymeric compounds that can change the handling of the fibres by protecting, aligning and modifying their wettability [34]. Besides protecting the fibres from breaking and improving the handling of the carbon filaments, the sizing has the function of acting on the fibre/matrix adhesion properties influencing the performance of the final composite [35]. However, during the pyrolysis process, much of the sizing is degraded, along with the polymer resin, resulting in the loss of its functionality as an interface between the fibres and the matrix. The residual material observed in Figure 3 is not considered to be a negative aspect, since the pyrolysis process was controlled so that no degradation of the fibre was observed. Rodrigues et al. [12] observed that the pyrolysis process carried out on carbon fibre at temperatures of 450 °C and 500 °C for 3 h does not cause damage to the carbon fibre surface, being the best temperatures for carrying out the carbon fibre recycling process, but it is possible to observe sizing residues on the fibre surface. As the pyrolysis process in this work was carried out at 500 °C for 4 h, the extent of the sizing of the fibres pyrolysed during this study was smaller compared to the results obtained by Rodrigues [12].

### 3.2. Density and Void Content

After the manufacturing process of cCFRP and rCFRP, the density values of these specimens were obtained following the Archimedes method according to ASTM D792-08 Procedure A [29]. The density values of the specimens cCFRP and rCFRP were obtained, and they are shown in Table 3. 

Using the density values analysis, it is possible to observe that the density values for the cCFRP and rCFRP were very close to those provided by the cured prepreg supplier (1.57 g/cm^3^ [25]).

This difference between the composites produced and the data provided by the suppliers is mainly due to the manufacturing process adopted (manual lamination and vacuum bagging) and the difference in the fibre volumetric fraction of the composite. It is possible to reach the maximum value of 50% (55.29% [25]) of the fibre volume fraction and the void content observed in the test specimens, which is also determined by the manufacturing process adopted. As a second method, the helium pycnometry method was used for measuring the composites’ density and the density of their constituents: the Araldite^®^ LY 5052/Aradur^TM^ 5052 resin system after the curing process, cCFRP and rCFRP. The results are presented in Table 4.

The average density values obtained using the helium pycnometer method for composites produced with both commercial CF and recycled CF compared with the average values obtained via the Archimedes method showed a difference of only 1%, which demonstrated that the value found in both assays is reliable. The density values for the resin and the commercial CF compared to the technical data provided by the manufacturer also proved to be consistent, and the density value for the recycled fibre showed a slight change compared to the commercial CF, a factor associated with probable moisture absorption fibres due to the loss of sizing after the pyrolysis process.

### 3.3. Mechanical Tests

#### 3.3.1. Flexural Tests

From the flexural test, it was possible to obtain the values of modulus of elasticity in bending ($E_{B})$, maximum stress ($\sigma_{fM})$, strain at the maximum stress ($\varepsilon_{fM})$ and maximum strain ($\varepsilon_{ff})$ for the specimens tested with both recycled and commercial fibres. For a quantitative comparison between the values obtained for the specimens of rCFRP in relation to the specimens of cCFRP was used the following Equation (10), where $X_{r}$ is the rCFRP properties, $X_{c}$ is the cCFRP properties and $X_{\%}$ is the value of property loss analysed as a percentage [36].
(10)$$X_{\%}=\left(1-\left(\frac{X_{r}}{X_{c}}\right)\right)\cdot 100\%$$

The mean values and standard deviation of the modulus of elasticity in bending, maximum stress and strain at the maximum stress for the specimens, as well as the value of the loss in each analysed property as a percentage, are summarised in Table 5.

Figure 4 makes it possible to observe the stress versus strain curves of all flexural test specimens. From the data obtained via the mechanical tests, as can be seen in Figure 4, it was possible to observe the reduction in the mechanical properties in the composites produced, with recycled carbon fibre being the highest reduction value for the strain at maximum stress, around 17.4%. Moreover, the rCFRP had a reduction of around 17.3% in relation to the E_B_ and a reduction of around 13.2% in relation to the maximum stress. The reduction in the mechanical properties was expected in rCFRP. It can, therefore, be guaranteed that composites made with rCRFP will have lower values for all their mechanical properties compared to composites made with commercial fibres. The decrease in E_B_ may be related to the loss of sizing and the consequent decrease in fibre-adhesive matrix interaction, which would lead to a decrease in the interfacial shear strength of the matrix [37]. Furthermore, the chosen manufacturing method may have introduced defects, such as fibre misalignment and the number of voids present in the specimens, as can be seen in Figure 5, which shows that the values found are realistic, but the manufacturing method used needs to be optimised to obtain specimens with better mechanical properties.

Figure 5 shows images generated via the SEM of the fracture region rCFRP and cCFRP specimens to compare the fracture behaviours that occurred in both specimens. The fracture analysis showed that the rCFRP specimen in Figure 5a presented a mix of fracture failure through the cross-sectional area, and in Figure 5b, it can be noted the cCFRP specimen only presented fracture failure on the upper and lower layers; we did not observe via the SEM technique any failure aspect in the centre of the specimen.

From the fracture aspects analyses, a cross-sectional failure can be observed in rCFRP, where it can be a sequence of failures in the specimen cross-section or a mixture of translaminar and interlaminar failure processes. According to Greenhalgh [38], a failure of a laminated composite can be divided into three types: the interlaminar type occurs when in the laminate plane, in which the layers have separated; the translaminar type in which fibres have been broken through the thickness; and the intralaminar type through the thickness in which only the matrix or fibre/matrix interfaces have been broken when the ply splits.

The presence of voids promoted the interlaminar crack propagation in the void directions, and translaminar failure through the thickness was promoted by the bending loading [39], indicating that mode II (shear) was the dominant failure [38,40].

The cCFRP specimens exhibited interlaminar failure on the upper and lower laminate surfaces during flexural tests, but in the upper layer, the interlaminar cracks were predominant, and a mixture of failure processes in the lower layer, interlaminar and translaminar cracks can be observed due to the maximum bending load localised in this region [41]. It is not possible to observe the intralaminar crack in flexural specimens because this failure type is generated at the interfaces between adjacent plies [38,40].

The absence of sizing caused by the pyrolysis process favours a poor interface between the fibre and matrix adhesion in rCFRP, and the void content generated via the manufacturing process contributed to the lower values of mechanical properties in both composites, rCFRP and cCFRP [42]. Therefore, it is possible to assume that the pyrolysis process used was efficient and generated the expected reduction in the mechanical properties of the rCFRP because of the sizing degradation, but as there was no superficial damage to the carbon fibre, it is possible to use it in a new application.

#### 3.3.2. Tensile Test

The data obtained via the tensile test are similarly organised to fit the results of the flexural test, as can be seen in Table 6. Table 6 shows the mean values and standard deviation of Young’s modulus (E), the maximum stress ($\sigma_{M}$), the strain at the maximum stress ($\varepsilon_{M}$) for the specimens tested with both recycled and commercial fibres and the value of the loss in each of the analysed properties as a percentage. Figure 6 shows the stress versus strain curves of all the specimens tested. The results obtained from the tensile test present the same behaviour as those obtained from the flexural test, namely the reduction in all the mechanical properties for the rCFRP specimens.

The values for the maximum stress were around 36.02%, and for the strain at the maximum stress, values of around 30.22% were relatively higher than the values presented in the flexural test, demonstrating that for purely tensile loads, the rCFRP are significantly affected. However, the reduction of about 17.75% in relation to the modulus of elasticity was similar to the reduction shown in the flexural test; this means that the fibres’ capacity to deform elastically remained the same, while deformation and rupture were proportionally smaller when the values obtained for the specimens manufactured with commercial carbon fibres were compared with the technical data values for the commercial prepreg. Anyway, it must be taken into account that the resin contained in the prepreg (Epoxy Resin 8552, HexPly^TM^, Salt Lake City, UT, USA) was different from the resin system used in the manufacturing of the Araldite^®^ LY 5052/AradurTM 5052 (Huntsman^TM^, Basel, Switzerland) test specimens, with this being one of the factors that contribute to the difference between the values. Table 6 shows the technical data of the prepreg provided by the manufacturer (HexPly^TM^, Salt Lake City, UT, USA).

Alves et al. [43] subjected the recycled composite to tensile tests via pyrolysis and observed the same behaviour reported here. They observed that the tensile modulus was very close for the two laminates. On the other hand, the tensile strength and ultimate longitudinal strain are about 40% lower in the recycled specimens. The loss of tensile strength at this point may be attributed to a combination of the damage on the surface of the fibre shown via micrography and the voids in the resin-rich areas. Most of the tensile load carried by a carbon fibre is transmitted in the fibre surface; any disruption to the surface can result in a mass-disproportionate loss of tensile properties. It was concluded that this reduction in shear stress and tensile strength is due to the weak interaction between the matrix and the resin because of the sizing loss.

Feraboli et al. [36] studied the mechanical properties of the recycled composite obtained via a chemical process, and they observed decreases in tensile strength flexural of approximately 25–30% and flexure strength of approximately 70% and 52% compared with results obtained for those manufactured with commercial carbon fibre.

According to Pimenta and Pinho [6,7], the pyrolysis process is a good recycling process because it has a high retention of mechanical properties, the potential to recover chemical feedstock from the resin and no use of chemical solvents. They also observed a decrease in mechanical performance of approximately 20% for the elastic properties of the recycled composite.

In addition to these factors, another issue that can affect the reduction in the modulus values is the alignment of the fibres in the final composite. As mentioned above, the pyrolysis process of carbon fibres removes the matrix and sizing, both of which are important for maintaining fibre alignment. After the pyrolysis process, the carbon fibres removed from the oven have no surface protection and can cause breakage and misalignment. This misalignment can be exacerbated by the manufacturing process chosen, which, in this case, is the hand layup method. This method consists of impregnating the fibre surface with resin using a spatula or impregnation roller. The back-and-forth movement of the object can cause some misalignment of the fibres. This factor drastically affects the tensile modulus results; van de Werken et al. [44] found that samples with sizing but no alignment had a 35% lower modulus than those without sizing but with alignment. In the model described by van de Werken et al. [44], the composite modulus increases monotonically as the fibres become more aligned along the tensile axis. Turner et al. [45] demonstrated in their studies that fibre alignment is a critical factor for attaining high mechanical properties and high recovered fibre utilisation.

Figure 7 shows images generated via SEM for the fracture region of one of the rCFRP and cCRFP to examine the difference between the fracture behaviours of these two materials during tensile tests. The fracture analysis of the rCFRP specimen showed a typical tensile fracture failure, with the translaminar failure being represented by the cross-sectional failure of the specimen (Figure 7a) that occurred when the fibres broke during tensile loading (Figure 7b) [37,38]. This dominant process is associated with the through-thickness tow, which tends to debond, leading to the development of local matrix cracks around the tow, represented by a higher number of fibre detachment marks (Figure 7b).

Figure 7b also presents an interlaminar crack in the region with detachment marks between the fibre and matrix promoted by the fracture at the fibre/matrix plane interface. During the pyrolysis process, the sizing, which is usually presented as a solution or an emulsion consisting of polymeric components [34], also degraded at 500 °C, causing a reduction in the adhesion properties between the fibre and the matrix, decreasing the mechanical properties of rCFRP compared to those made with commercial fibres that contain sizing. The intralaminar failure is represented by the fibre and matrix region on the right side of Figure 7b.

Another failure aspect can be observed in Figure 7, i.e., a tensile fracture aspect, and the fibre pullout (Figure 7a) is represented by a circular hole in the matrix (Figure 7b). Using the manufacturing process, we identify voids on the matrix surface (Figure 7b). Inadequate distribution of the matrix in the composite layers encourages voids to form. These voids promote discontinuity in the distribution of forces in the composite and become a point of stress accumulation. Stress accumulation can increase crack propagation energy and promote delamination in the intra- and interlaminar regions of composites. If there is an interaction between the rearrangement of stresses along the void and the misalignment of the fibres, it can lead to fibre/matrix debonding and the local overloading of the fibres, followed by a progressive decrease in the stability of the fibres, promoting their rupture and leading to translaminar failure [42].

Figure 7c presents a mixture of interlaminar, intralaminar and translaminar fracture failure through the cross-sectional area of the cCFRP specimen, as reported by Greenhalgh [38]. The intralaminar failure can be observed in the upper side of Figure 7c, where the cracks propagate on the fibre resin/matrix layer, splitting it in the plane. The lower side of Figure 7c showed translaminar and interlaminar failure. The first one is represented by the cross-sectional crack, and the last one is represented by the parts of the layers in the 0° direction that appear in the centre of the translaminar crack.

Figure 7d shows the interlaminar failure in magnification mode. It is shown that a good impregnation of the matrix in the fibre’s surface (Figure 7d region A) resulted from a strong fibre matrix interface due to the presence of a sizing compound on the commercial carbon fibre surface. The tensile fracture aspect, fibre pullout and broken fibres can also be observed in region B of Figure 7d.

Based on these results, the mechanical properties of the composites were estimated using Equations (3) and (4). The estimated modulus of elasticity is presented in Table 7, where the estimated values for composites produced with commercial or recycled CF are compared with the results obtained using the mechanical test presented in Table 6; the estimated values obtained are 57.6% lower for commercial carbon fibres and 53% lower for recycled carbon fibres. These differences are mainly due to the fact that the number of voids is not taken into account in the calculations to estimate the mechanical properties of the composites, which, depending on the process used, can represent up to 4% of the void content [46,47,48], which can reduce the properties of the composites by more than 20% [47,48,49], as well as possible defects related to the composite manufacturing process, which can cause early failure and lower values of the modulus of elasticity.

#### 3.3.3. Mode I Fracture Toughness Characterization

To characterise the mode I fracture toughness of rCFRP specimens manufactured via the HLUP + VB process, the DCB testing method was used according to ASTMD5528-01 [31]. Of the five specimens tested, only four rCFRP specimens presented valid results.

In Figure 8, it can be seen that the load increases linearly until it reaches the maximum load where the crack starts after a gradual decrease due to crack propagation. The crack propagates in the longitudinal direction of the composite, i.e., in the direction of the warp, and when the front of the crack tip encounters a physical barrier, which may be a resin-rich region, a void, a fibre misalignment, or even, depending on the fabric architecture, an artificial high toughness is induced until this energy is overcome and crack propagation continues [48,49]; then, ‘stick-slip’ behaviour is observed. Due to these local variations in composite systems, particularly in the fabric architecture, here is the plane wave architecture, which is where this phenomenon is seen.

Comparing the G_I_ values (Table 8 and Figure 8c) for the rCFRP specimens manufactured via the HLUP +VB process with those found in the literature [50], it can be observed that rCFRP specimens have approximate G_I_ values. VaRTM samples are the best samples for comparison as they can show void formation inside and on the surface as no external pressure is used in this process, as well as samples obtained by hand lay-up + vacuum bagging (HLUP + VB) as used in this study.

### 3.4. Sand Ladder Platform Prototype

#### 3.4.1. Sand Ladder Platform Prototype Dimensioning

Another result obtained from the density data generated was the estimation of the mass of the sand ladder platform product using Equation (2) that was generated using the Solid Edge 2019 software (version 2019, Siemens, Munich, Germany). The value found for the estimated mass of the product is equal to 3.56 kg, and when compared to the mass of other similar products, it proved to be practical and within reality. As an example, it is possible to mention the sand ladder platforms acquired from the LITE-WAY brand [51] with approximately 2.66 kg and the SUNPIE brand with approximately 6.1 kg.

Based on data collected from market research and taking into consideration the fact that the sand ladder platform product must have minimum dimensions to be functional, a minimum value of 800 mm and a minimum value of 250 mm were adopted as length restrictions. These values represent the minimum dimension requirements adopted during the project.

Another requirement established was the maximum tension that can be supported by the material. The average value for this requirement was generated from the average of the maximum flexural stress results obtained by testing specimens with recycled carbon fibres, disregarding the value with the greatest discrepancy to reduce the standard deviation, generating a result with greater reliability.

After establishing these requirements, the load suffered by the platform was calculated according to ASTM D790-10 [27], and the product was dimensioned by following the established requirements. Table 9 presents the values found for the load suffered by the sand ladder platform, the established dimensions, the maximum stress in service and the safety factor adopted.

The values obtained when sizing the sand ladder platform had, as final dimensions, values close to the minimum values of similar products, with the focus being on reducing material consumption and reducing the maximum tension suffered by it. The relationship between the maximum stress experienced by the product and the rupture stress of the rCFRP generated a safety factor close to the established minimum factor of 3.15 and provided good reliability for the product’s operation. Furthermore, the calculated maximum deformation was 70% lower than the maximum deformation suffered by the composites in the flexural test with recycled CF, which corroborates with the sizing step carried out.

#### 3.4.2. Lamination Mould

The mould employed for laminating the sand ladder platform sections manufactured using additive manufacturing required a manufacturing time of approximately 28 h, with 7 h of manufacturing time for each section. The adopted process proved to be adequate, generating a product with good dimensional accuracy, a rough lamination surface and a precise fit.

The choice not to adopt a surface treatment for the lamination surface aimed to generate intentional roughness on the surface of the sand ladder platform to generate better traction for car tires during the use of the product. Figure 9 illustrates a section of the mould, representing the junction between the sections by using wooden dowels and the mould in its usage configuration.

To manufacture the system for applying the vacuum, a total time of 7 h 30 min was required, with 1 h 30 min for each section. Figure 9d,e shows the already assembled vacuum system and its positioning on the mould.

During the manufacturing of a section of the sand ladder platform product, damage occurred to the structure designed for vacuum application during its removal from the mould surface due to the high adhesion of the sealing adhesive placed between the two substrates and the low strength of the material used to make the system, which made it impossible to reuse it to make another section. However, the system fulfilled its function by facilitating the application of intermediate vacuums and reducing the use of consumables (sealing adhesive and vacuum bag); this demonstrated that the choice of material used for manufacturing the system must be rethought.

The mould used proved to be suitable for the production process, generating a product with good dimensional accuracy and surface quality, as well as an easy mould release process. However, due to the temperature used in the post-curing process, the mould presented deformations in its outer layers and the lamination surface. Such deformations do not compromise the reuse of the mould to manufacture a new section but demonstrate that the choice of material used for its manufacturing must be rethought.

#### 3.4.3. Sand Ladder Platform Manufacturing and Testing

The production process used to manufacture a section of the sand ladder platform product proved to be adequate in terms of its final quality but expensive due to the manufacturing time and quantity of material used. Figure 10 shows the characteristics of the product still in the mould, immediately after demoulding and after the finishing process.

After the finishing process of the section, it had a mass of 574.88 g and a lateral dimension of 165 mm, longitudinal dimension of 255 mm, maximum thickness of 12 mm and minimum thickness of 6 mm. If the other sections to be manufactured had the same dimensions, the sand ladder platform would have an approximate mass of 2.87 kg and a length of 825 mm.

The estimated mass value of the product calculated was equal to 3.56 kg (24% higher than the mass estimate from the section due to the inclusion of products used in the finishing process), demonstrating that this value is close to reality; however, the design dimensions of the section produced were 15 mm smaller longitudinally and 10 mm smaller laterally (due to wear in the finishing process), bringing the product closer to the minimum dimensional requirements established in this study.

After the finishing process, field testing was carried out on the produced section. This did not suffer any damage after the test and proved the viability of the product. Figure 11 shows images taken at the moment the Ecosport Freestyle 1.6 passes over the sand ladder platform, and it remains on the platform surface for 10 min. According to the sand ladder platform calculations, the product would support a load of 4110.39 N (Table 9), which is approximately 420 kg. The car used weighs approximately 433 kg (Table 2), so it can be concluded that the product is perfectly designed. A visual inspection was performed on the prototype after the test, and no apparent damage was observed on the product.

## 4. Discussion

The use of composite materials is increasing day by day in various industries, and the aerospace industry is one of the main contributors to this growth. However, the losses incurred during the process, the existence of legislation regulating the disposal of this type of material and the high added value associated with the cost of the raw materials used have led to the search for recycling processes and the development of products made from recycled raw materials. However, before use, the properties of the recycled material must be investigated in order to determine its suitability for use in other product forms.

Scanning electron microscopy was used to evaluate the recycled carbon fibres obtained from the pyrolysis process. The surface images of the recycled fibres showed that the process used was efficient and produced recycled fibres free from damage but with some remnants of the original polymer matrix. The compound known as sizing was degraded during the pyrolysis process. It acts as a surface protection for the carbon fibre and also as a binder between the fibre and the matrix, playing an important role as a strong interface that distributes stress during mechanical stress. The data obtained in the mechanical tests confirmed this when the mechanical properties of rCFRP and cCFRP were compared, with the most notable reduction being in the tensile properties.

The density analysis showed that the values obtained via both methods, the helium pycnometer method and the Archimedes method, as well as for both compounds studied here, showed a difference of only 1%, demonstrating that the methods tested are reliable. The density values compared with the technical data provided by the manufacturer were found to be consistent, and the density value for rCFRP showed a slight change when compared with cCFRP, a factor related to the probable moisture absorption of the fibres due to the loss of sizing after the pyrolysis process.

Flexural and tensile tests carried out on specimens of cCFRP and rCFRP showed that the rCFRP presented a reduction in mechanical properties compared to those made with commercial carbon fibres. In both tests, a reduction of about 17% in relation to the modulus of elasticity was observed, indicating that this parameter was affected in the same way. However, the values for the reduction in the maximum stress are around 36%, and for the strain at the maximum stress, they are around 30%. The last two parameters were roughly twice higher than the values presented in the flexural test (12% and 17%, respectively), demonstrating that, for purely tensile loads, the rCFRP are significantly affected. This reduction is related to the loss of sizing during the pyrolysis process, resin residues, superficial damage to the fibres and some misalignments caused by their disposal, but they still have better potential for secondary applications than those initially designed, where high strength is required in combination with low density.

Possible surface treatment of the fibres after the pyrolysis process can be recommended if necessary to improve the mechanical properties, as this would result in an improvement in the fibre–matrix adhesive interaction lost due to sizing degradation, but the type of surface treatment chosen, process costs and viability on a large scale must be considered.

Furthermore, from comparison with technical data provided by the manufacturers of the raw materials used and from calculations used to estimate the mechanical properties, it can be observed that the manufacturing process needs to be improved to produce composites with fewer defects and, consequently, better mechanical properties.

The specimens subjected to the flexural tests showed a mixture of fracture aspects during the SEM analysis, but the rCFRP specimens showed more fracture aspects after the mechanical tests. This is mainly due to the presence of voids in the rCFRP produced during the manufacturing process, which promoted crack propagation as interlaminar, and through the thickness of the specimen. In the flexural tests, the void content interfered more with the mechanical results than the sizing factor. These voids promote discontinuity in the distribution of forces in the composite and become a point of stress accumulation that can promote delamination in the intra- and interlaminar regions of composites, and if there is an interaction between the misalignment of the fibres, this can lead to translaminar failure.

More from the fractographic analyses, both composites subjected to tensile tests presented the three types of failure process: translaminar, interlaminar and intralaminar. The rCFRP was completely fractured throughout the cross-sectional area, and the cCFRP was partially fractured, presenting the mixture of failure processes on the specimen surface, some intralaminar cracks in the top layer, a translaminar crack in the bottom of the specimen and an interlaminar crack inside the translaminar crack. The partial failure of the cCFRP indicates that the presence of sizing is a critical factor in the tensile strength of the composite, as the sizing promotes a strong matrix/fibre adhesion interface that distributes tensile stresses throughout the composite, thereby improving the mechanical properties of the composite when subjected to tensile loads. This is why the mechanical values between rCFRP and cCFRP in tensile tests are so different.

To complete the mechanical characterisation of rCFRP and validate the manufacturing process of the recycled carbon fibre composite, some specimens were produced via the HLUP + VB process and characterised according to ASTMD5528-01 [31]. When compared with the literature, the results showed approximate and satisfactory values for composites used in aerospace structures produced via the VaRTM process. Therefore, rCFRP can be used to produce the prototype of the sand ladder platform.

The dimensioning of the recycled carbon fibre sand ladder platform product proved to be viable, given the data obtained from the mechanical tests carried out and the comparison with similar existing products. In addition, the design of the mould, its manufacturing and the final characteristics showed excellent properties and practicality of use; however, the damage to the mould structures and the vacuum application system demonstrated the need to reconsider the constituent material of these two structures.

The manufactured section of the sand ladder platform can withstand the conditions of use for which it was designed, but the conditions of the manufacturing process need to be optimised to obtain a product with characteristics that are more faithful to the design and of better final quality. This requires new studies of the manufacturing process and product design. During the field test, the prototype did not present as damaged or with flexural failure.

## 5. Conclusions

In this work, an extensive study was carried out for the recycling of carbon fibres, as well as the production of a new composite and its mechanical and fractographic characterisation, for subsequent application in the development of a prototype of a sand ladder platform. The main points observed during this study are highlighted below.

The SEM images indicate that the pyrolysis process was efficient. The carbon fibre surfaces did not present any degradation site. The stretch marks visible on the carbon fibre surface are attributed to the extrusion process to which they are subjected during the manufacturing process.

Comparing the physical properties, it is observed that the composite density values are close, even though the void content values in the recycled composite are higher compared with the commercial composite.

The results obtained from mechanical tests showed a higher mechanical loss of rCFRP compared to those made of commercial fibre. The sizing degradation during the pyrolysis process affects the results of the tensile tests more than those for the flexural tests, indicating the importance of the good interface between the fibre matrix that promotes a strong matrix/fibre adhesion interface and distributes tensile stresses throughout the composite, thereby improving the mechanical properties of the composite when subjected to tensile loads. The cCFRP did not completely break during the tensile test, indicating the importance of a strong fibre/matrix interface.

In the flexural tests, the void content interfered more with the mechanical results than the sizing factor. The voids promoted stress accumulation, increasing the crack propagation energy and promoting two types of delamination processes: inter- and intralaminar. When this interaction occurred with the misalignment of the fibres, the translaminar was observed.

The fracture toughness characterisation showed that the fCFRP specimens exceeded the approximate values obtained via the VARTM process for composites used in aerospace structures.

The steps and methods chosen to design, manufacture and test the prototype of the recycled carbon fibre sand ladder platform were appropriate and gave satisfactory results. The prototype proved to be of excellent final quality as it had high mechanical strength to bending and was easy to use. Only a few adjustments to the mould used to manufacture this product should be considered to obtain a final product that can be sold.

Therefore, it is possible to assume that the pyrolysis process adopted was efficient at generating an expected decrease in mechanical strength for the recycled composite, which shows that it is still possible to use these materials for different applications than those initially designed.

## Figures and Tables

**Figure 1 materials-17-00423-f001:**
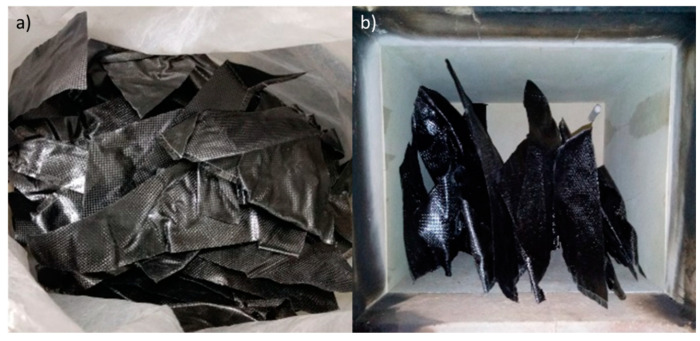
Scraps of prepreg 8552/AS4 plain-wave (Hexcel) (**a**) as received and (**b**) as placed into the furnace.

**Figure 2 materials-17-00423-f002:**
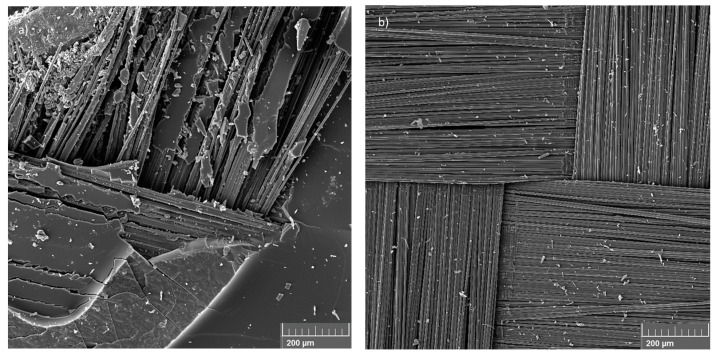
Prepreg scrap surface: (**a**) before and (**b**) after the pyrolysis process (with a magnification of 200×).

**Figure 3 materials-17-00423-f003:**
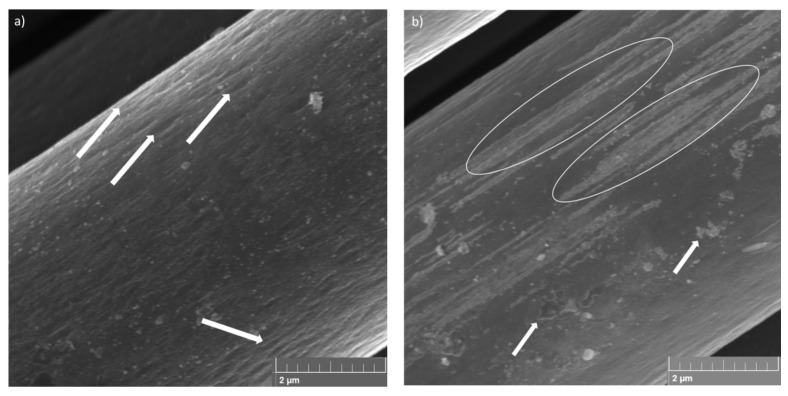
Surface of recycled fibres: (**a**) stretch marks and (**b**) sizing and polymeric matrix residues (with a magnification of 40,000×). (Circles identify the stretching marks, and Arrows the residues of the polymeric matrix).

**Figure 4 materials-17-00423-f004:**
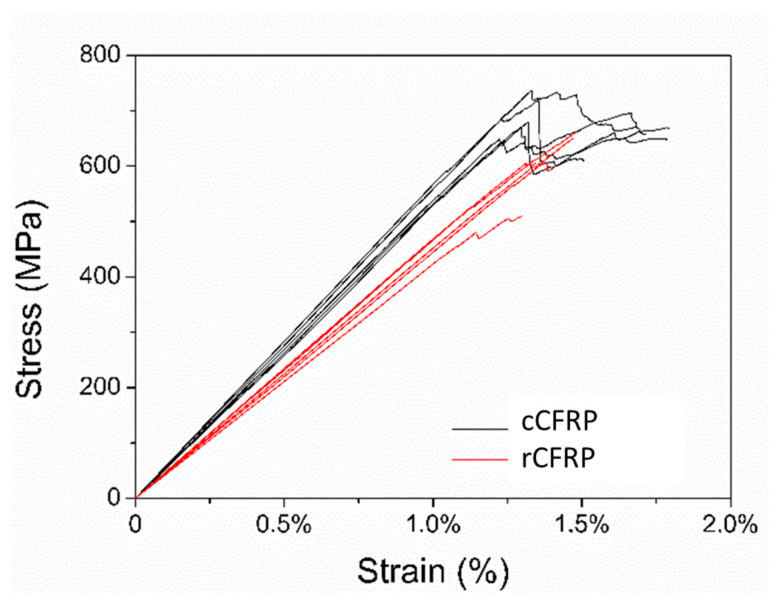
Stress versus strain plot for all flexural test specimens.

**Figure 5 materials-17-00423-f005:**
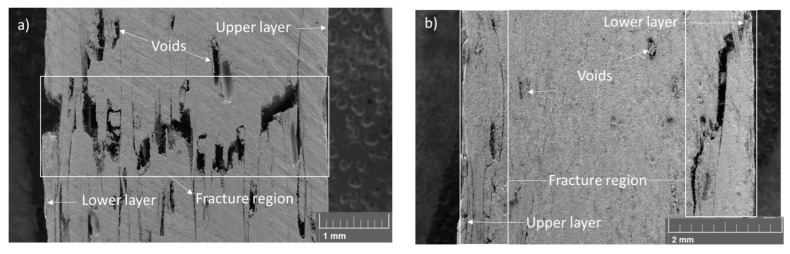
Fracture surfaces of the specimens (**a**) rCFRP and (**b**) cCFRP (with a magnification of 50×).

**Figure 6 materials-17-00423-f006:**
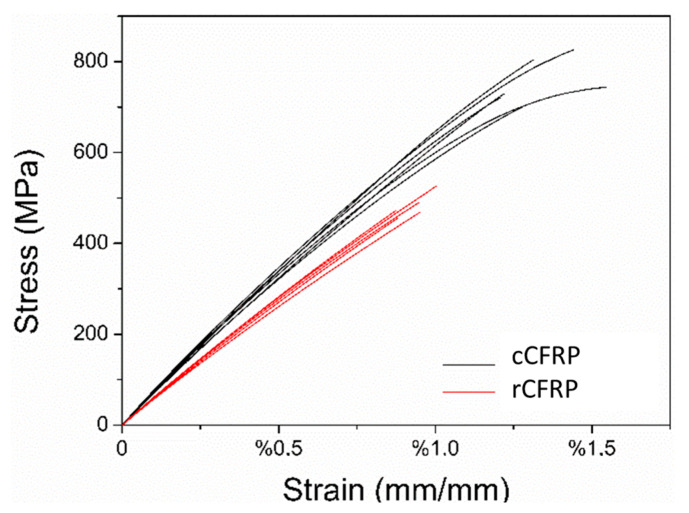
Stress versus strain plots for all tensile test specimens.

**Figure 7 materials-17-00423-f007:**
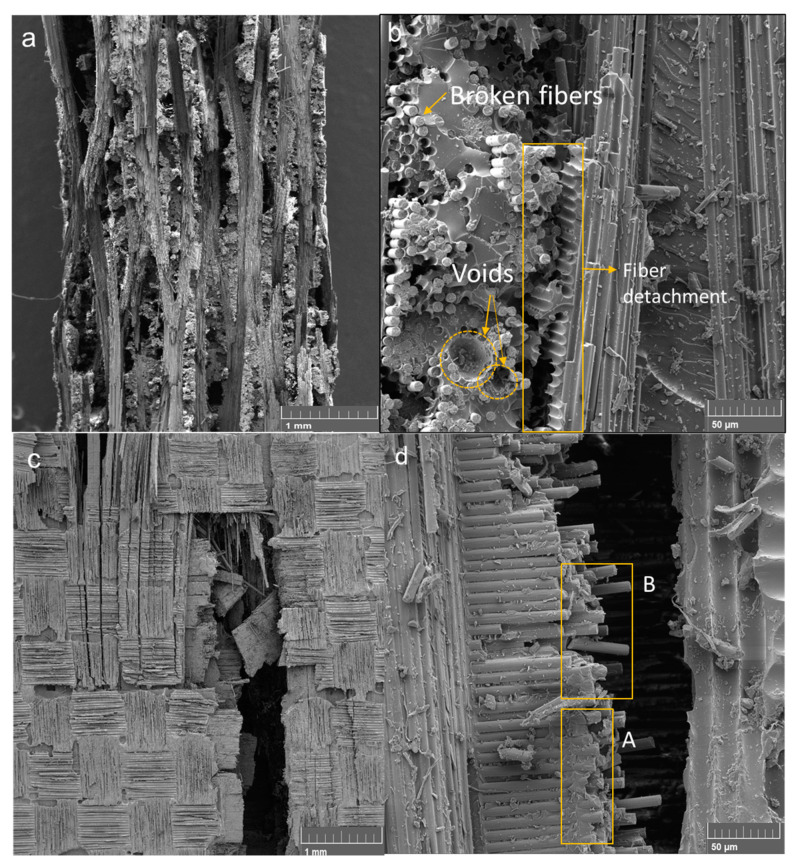
The fracture surfaces of rCFRP specimens with magnifications of (**a**) 50× and (**b**) 1000× and cCFRP specimens with magnifications of (**c**) 50× and (**d**) 1000×. A: Good impregnation of the matrix in the fibre’s surface; B: fibre pullout and broken fibres.

**Figure 8 materials-17-00423-f008:**
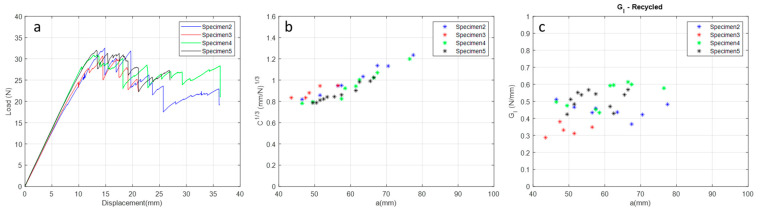
Mode I results: (**a**) Curve P versus δ; (**b**) $C^{1/3}$ versus delamination extension a e; (**c**) Curve G_I_ versus the delamination extension a of rCFRP specimen manufactured via the HLUP + VB process and analyzed at 25 °C.

**Figure 9 materials-17-00423-f009:**
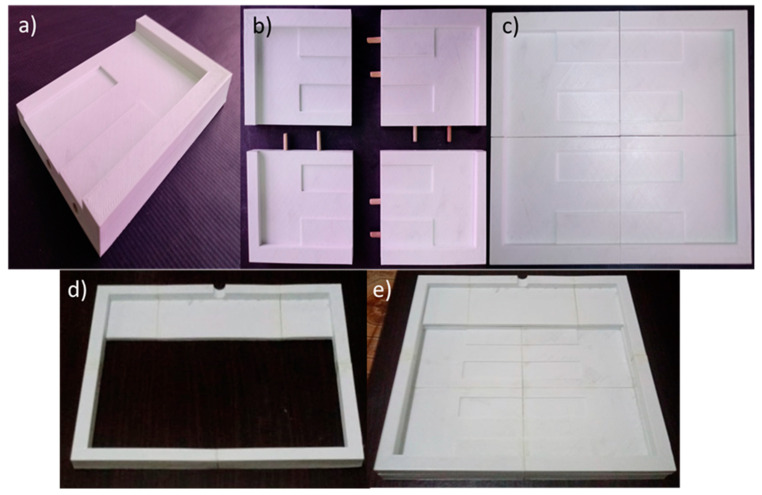
(**a**) Mould section, (**b**) junction between sections, (**c**) mould in-use configuration, (**d**) assembled vacuum system and (**e**) vacuum system positioned on the mould.

**Figure 10 materials-17-00423-f010:**
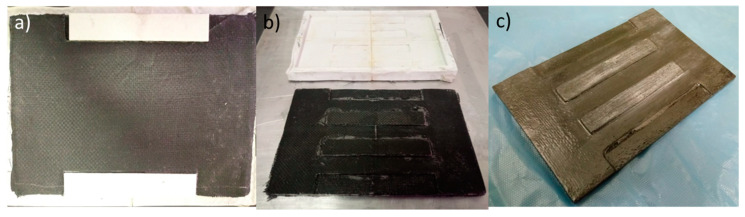
Prototype feature: (**a**) about the mould; (**b**) immediately after demoulding; (**c**) after finishing.

**Figure 11 materials-17-00423-f011:**
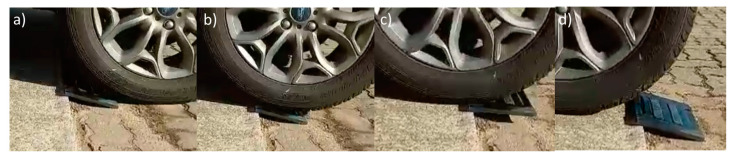
Field test: (**a**) initial moment; (**b**,**c**) passage of the car over the section; (**d**) end of passage.

**Table 1 materials-17-00423-t001:** Laminate and specimen dimensions, stack sequence and ASTM standards used in the mechanical tests.

	Flexural Test	Tensile Test
Stack Sequence	[(0/90)]_22_	[(0/90)]_13_
Laminate plate dimensions	200 mm × 150 (mm)	270 mm × 200 (mm)
ASTM standard	ASTM D790-10 [27]	ASTM D3039M-08 [28]
Specimens dimensions	90 × 15 × 4 (mm)	250 × 25 × 2.5 (mm)

**Table 2 materials-17-00423-t002:** Dimensions of the Ford EcoSport Freestyle 1.6 [32].

Parameters	Values
Length	4241 mm
Length between the axis	2521 mm
Front gauge	1524 mm
Trunk	362 L
Weight	1243 kg
Useful load	433 kg

**Table 3 materials-17-00423-t003:** Density results obtained via the Archimedes method according to ASTM D792-08 Procedure A [29].

Density (g/cm^3^)	Average Values ± Standard Deviation
cCFRP	1.46 ± 0.06
rCFRP	1.46 ± 0.01

**Table 4 materials-17-00423-t004:** Density results obtained via the helium pycnometer method.

Density (g/cm^3^)	Average Values ± Standard Deviation
cCFRP	1.48 ± 0.001
rCFRP	1.47 ± 0.001
* Commercial fabric	1.75 ± 0.001
Recycled fabric	1.83 ± 0.001
** Resin system	1.16 ± 0.000

* Texiglass^TM^, Vinhedo, São Paulo, Brazil; ** HexPly^TM^, Salt Lake City, UT, USA.

**Table 5 materials-17-00423-t005:** Summary of the data obtained via the flexural test.

	$E_{B}$ (GPa)	$\sigma_{fm}$ (MPa)	$\varepsilon_{fM}$ (%)	$\varepsilon_{ff}$ (%)
cCFRP	54.4 ± 2.0	699.6 ± 32.4	1.44 ± 0.18	1.7 ± 0.4
rCFRP	45.0 ± 1.9	607.0 ± 60.5	1.39 ± 0.09	1.4 ± 0.1
X (%)	17.3	13.2	3.76	17.4

**Table 6 materials-17-00423-t006:** Summary of the data obtained via the tensile tests.

	E (GPa)	$\sigma_{M}$ (MPa)	$\varepsilon_{M}$ (%)
cCFRP	67.16 ± 1.91	753.96 ± 49.79	1.33 ± 0.13
rCFRP	55.24 ± 1.74	482.41 ± 27.26	0.93 ± 0.05
X (%)	17.75	36.02	30.22
AGP193-PW [25] *	68	828	-

* AGP193-PW [25] (HexPly^TM^, Salt Lake City, UT, USA) Test condition: T (25 °C), fibre direction at 0°.

**Table 7 materials-17-00423-t007:** Estimated mechanical properties.

Composite	Volumetric Fraction (%)	Estimated Modulus of Elasticity (GPa)
cCFRP	54	126.0
rCFRP	48	95.8

**Table 8 materials-17-00423-t008:** Mode I interlaminar fracture toughness values for the rCFRP and cCFRP specimens and comparisons with the literature [50].

Manufacturing Process	Mean G_I_ (N/mm)	Standard Deviation G_I_ (N/mm)
AS4/RMT6 VARTM [50]	0.420	0.070
rCFRP HLUP + VB	0.475	0.105

**Table 9 materials-17-00423-t009:** Sand ladder platform product dimensioning results.

Parameter	Values
Platform (N)	4110.39
σ_max_ (MPa)	199.81
$\varepsilon_{\max}$ (%)	0.44%
FS	3.15
Length (mm)	875
Width (mm)	270
Thickness (mm)	10

## Data Availability

The data shared is in accordance with consent provided by participants on the use of confidential data.

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
