# Peer review of "Mechanical Strength and Surface Analysis of a Composite Made from Recycled Carbon Fibre Obtained via the Pyrolysis Process for Reuse in the Manufacture of New Composites"

_materials, 2024, doi:10.3390/ma17020423_

Round 1

Reviewer 1 Report

Comments and Suggestions for Authors

This paper analysis the possible reuse of carbon fibers reiforced composites to produce new products specifically "Sand ladder Platforms". The experimental characterization, discussion and final production of the prototype have been correctly completed. Maybe the final application is not so extensively applied so at the end of the discussion the authors may propose other possible items that could be made with the same process in order to extend the overall interest in their paper. 

Few points of discussion:

Page 5 line 184 The authirs should discuss in more details the selected equation (3) and the selcted values of Ef and Em

Page 6 line 317 The presence of sizing residuals. It is not clear whether these residuals are to be considered as a negative aspect or not. Could the pyrolizing process could be improved to completely eliminate them? From the micrograph there seem to be really a negligible damage to the surfaces. So the paragraph on lines 432-436 could be rephrased. (see also 448-449)

Line 549 Please rephrase this paragraph: it is not clear.

Discussion 

The authors hint to a possible resizing of the recycled fibers: a rough extimation of the cost? 

Comments on the Quality of English Language

Some minor changes should be considered

Author Response

Comments 1: Page 5 line 184 The authors should discuss in more detail the selected equation (3) and the selected values of Ef and Em.

Response 1: Thank you for pointing this out. The changes can be found on page 5, line 184 to 189, highlighted in red.

where Ee is the estimated modulus of elasticity; Ef is the modulus of elasticity of the fibre, this value was taken for commercial fibre from the technical data provided by the manufacturer (AS4 3K 200g/m² /HexPlyTM), and for recycled fibre, the same value was multiplied by the resistance loss in modulus obtained from the flexural test results); vf is the volume fraction of the fibre, and Em is the modulus of elasticity of the matrix (Epoxy resin 8552) according to the technical data provided by the manufacturer

Comments 2: Page 6 line 317 The presence of sizing residuals. It is not clear whether these residuals are to be considered as a negative aspect or not. Could the pyrolyzing process could be improved to completely eliminate them? From the micrograph, there seems to be really negligible damage to the surfaces. So, the paragraph on lines 432-436 could be rephrased. (see also 448-449).

Response 2: The changes can be found on page 8, lines 317 to 320, highlighted in red.

The sizing is a superficial treatment applied on the fibres which generally consists of polymeric compounds that can change the fibres handling by protecting, aligning and modifying their wettability [36]. Besides protecting the fibre from breaking and improving the handling of the carbon filaments, the sizing has the function of acting on the fibre/matrix adhesion properties influencing the performance of the final composite [37]. However, during the pyrolysis process, much of the sizing is degraded along with the polymer resin, resulting in the loss of its functionality as an interface between fibre and matrix. The residual material observed in Figure 3 is not considered a negative aspect, since the pyrolysis process was controlled so that no degradation of the fibre was observed. Rodrigues et al. [12] observed that the pyrolysis process carried out on carbon fibre at temperatures of 450 and 500ªC for 3h does not cause damage to the carbon fibre surface, being the best temperatures to carry out carbon fibre recycling process, but it is possible to observe sizing residues on the fibre surface. As the pyrolysis process in this work was carried out at 500ºC for 4h, the amount of sizing of fibres pyrolyzed during this study was smaller when compared to the results obtained by Rodrigues [12].

The pyrolysis process was adjusted to a temperature of 500ºC to remove the resin without damaging the fibre surface. Tests were carried out at higher and lower temperatures in relation to the temperature reported here in this work. Pyrolysis carried out at lower temperatures resulted in excess resin on the fibre surface and at higher temperatures there were excess defects such as holes on the fiber surface. These results will be published in a future article.

The changes can be found on page 13, lines 456 to 459, highlighted in red.

Most of the tensile load carried by a carbon fiber is transmitted in the fiber surface; any disruption to the surface can result in a mass-disproportionate loss of tensile properties. It was concluded that this reduction in shear stress and tensile strength is due to the weak interaction between the matrix and the resin because of the sizing loss.

Comments 3: Line 549 Please rephrase this paragraph: it is not clear.

Response 3: The changes can be found on page 16, lines 547 to 551, highlighted in red.

The estimated Modulus of Elasticity is presented in Table 8, when the estimated values for composites produced with commercial or recycled CF are compared with the results obtained by the mechanical test presented in Table 6, the estimated values obtained are 57.6% lower for fibre commercial carbon fibre and 53% lower for recycled carbon fiber.

Comments 4: Do the authors hint at a possible resizing of the recycled fibres: a rough estimation of the cost?

Response 4: The authors appreciate your suggestion, but both topics will be discussed in a future paper.

4. Response to Comments on the Quality of English Language

Point 1: Some minor changes should be considered

Response 1:  The English version of the text has been corrected.

Reviewer 2 Report

Comments and Suggestions for Authors

Subject-wise the paper is of interest; yet it is an application/case study of composites [partial] recycling. Hence topic-wise the work is not among those rather general interest in the field of Materials and might  be more appropriate for a composites journal; this doesnot exclude Materials as a potential forum but there is some issue here. Actually, emphasis on  the particular application (sand ladder) makes the work appearing of less general interest than it actually is.   A new more general  title is needed if the particle is to be considered for Materials (the same applied to some extent to the abstract).

The study from the Composites practice is a detailed and appropriate one.

Findings such as that seen in Fig, 5 suggest that the new composite material is not equivalent to the  virgin one as regards some [of the] important mechanical responses.

The fact that the loweirng of mechanical properties was expected  does not lead to the authors’ conclusion that ‘it is possible to use it in a new application’ (line 409); the same applies to the claims in lines 462-464, 760-763 etc.

Undoubtedly the new product is inferior to the virgin one and the reasons (or some of the reasons) for that are exposed (loss of sizing etc). The losses (as regards level strength and E etc) and changes (e.g. mode of failure) are such that the new material cannot be viewed as an acceptable variant of the virgin material though it might used for another application. Improvements as regards the processing are possible and some are described in the manuscript.

The loss of strength is not surprising. The (smaller) loss of modulus is not discussed/expalined in detail. He it might be noted that modulus corresponds to an initial slope and is not affected by most of the factors that contribute to the loss of strength. Hence, I expect some discussion here.

Comments on the Quality of English Language

English: awkward at places, e.g. Increase development, wich (= which), flexurE test

Author Response

Comments 1: Subject-wise the paper is of interest, yet it is an application/case study of composites [partial] recycling. Hence topic-wise the work is not among those rather general interest in the field of Materials and might be more appropriate for a composites journal; this does not exclude Materials as a potential forum but there is some issue here. Actually, emphasis on the particular application (sand ladder) makes the work appear of less general interest than it actually is. A new more general title is needed if the particle is to be considered for Materials (the same applied to some extent to the abstract).

Response 1: The authors appreciate the proposed suggestion. The changes can be found on page 1, lines 12 and 22, highlighted in red.

This work aims to obtain recycled carbon fibre and develop an application for this new type of material. The sand ladder platform was chosen to be developed.

This research was initiated through an approach with the Brazilian aeronautical industry and academia, in this agreement it was requested that in addition to producing recycled carbon fibre, products would also be produced from it. In addition to the sand ladder platform, structures were developed for application in model aircraft competition and brakes. Each of these studies was published in journals and conferences separately.

In general, articles published on the recycling of composite materials only study the material itself and do not address the possibility of applying the new material. This article, in addition to the scientific approach, brings a technological application of the new material. Within this approach, the proposed article fits perfectly into the special issue to which it was submitted: Polymers: From Waste to Potential Reuse.

Comments 2: Findings such as that seen in Fig, 5 suggest that the new composite material is not equivalent to the virgin one as regards some [of the] important mechanical responses. The fact that the lowering of mechanical properties was expected does not lead to the authors’ conclusion that ‘it is possible to use it in a new application’ (line 409); the same applies to the claims in lines 462-464, 760-763 etc.

Response 2: The authors appreciate the proposed suggestion. The statement 'it is possible to use it in a new application' means that despite the loss of mechanical resistance of the recycled composite material made from pyrolyzed carbon fibre, the values obtained from the mechanical tests shown in Tables 5 and 6 indicate the possibility to use it for a new application, especially when these values are compared with estimated values of the elasticity mode for the prototype as presented in Table 8.

The changes can be found on page 12, lines 410 to 417, highlighted in red.

The absence of sizing caused by the pyrolysis process favours a poor interface between fibre and matrix adhesion in rCFRP and the void content generated by the manufacturing process contributed to the lower values of the mechanical properties. Therefore, it is possible to assume that the pyrolysis process used was efficient and generated an expected reduction in the mechanical properties of the rCFRP because of the sizing degradation, but as there was no superficial damage to the carbon fibre being possible to use it in a new application.

Other paragraphs with the same sentence were removed from the text.

Comments 3: Undoubtedly the new product is inferior to the virgin one and the reasons (or some of the reasons) for that are exposed (loss of sizing etc). The losses (as regards level strength and E etc) and changes (e.g. mode of failure) are such that the new material cannot be viewed as an acceptable variant of the virgin material though it might be used for another application. Improvements as regards the processing are possible and some are described in the manuscript.

Response 3: The authors are grateful for the reviewer's comments.

Comments 4: The loss of strength is not surprising. The (smaller) loss of modulus is not discussed/explained in detail. It might be noted that modulus corresponds to an initial slope and is not affected by most of the factors that contribute to the loss of strength. Hence, I expect some discussion here.

Response 4: The authors appreciate the proposed suggestion. Some comments about the loss of modulus are discussed in the text on page 12, highlighted in red.

However, the reduction of about 17.75% in relation to the modulus of elasticity was similar to the reduction shown in the flexural test, this means that the fiber capacity to deform elastically remained the same, while deformation and rupture are proportionally smaller.

4. Response to Comments on the Quality of English Language

Point 1: Some minor changes should be considered

Response 1:  English: awkward at places, e.g. Increase development, wich (= which), flexurE test

In some industries, such as aerospace, automotive and sports, there has been an increase in the development of more efficient and complex composite materials.

The words “wich and flexurE” were corrected.

Reviewer 3 Report

Comments and Suggestions for Authors

The manuscript entitled “Mechanical strength and surface analysis of a composite made from recycled carbon fibre obtained by the pyrolysis process for the development of a “Sand Ladder Platform” product” presents analyses of the recycled and commercial carbon fibres combined with the same resin/hardener system. The obtained data were compared. The authors conduct many experiments, i.e., the flexure, tensile, and Mode I ones. The manuscript is well-prepared, step-by-step, leading a potential reader throughout the measurements, preparation process, and testing process. I think the manuscript can be published in the Journal after minor amendments.

1.      Page 5, Eqs. (1)-(3): Why did the authors use the cross sign to denote multiplication? Please compare with Eq. (6) – there is a point sign, as it should be. The cross sign is used for vector multiplication (then one gets a vector quantity!).

2.      Page 6 line 229 (and many others like this!): There is no “CF” in Eq. (4). There is “F_c” (“c” as the subscript) instead of.

3.      Page 6 lines 245-252: Why did the authors choose this type of car (EcoSport, Ford)? Is this surreptitious advertising? The authors can use any other car, i.e. (more or less), about the same length, the length between the axis, and the total weight.

4.      Page 9 Table 3: Typically, up to two significant digits are given for the error value.

5.      Page 10 Eq. (10): see point 1!

6.      Page 11 lines 380-381 (and many other in the manuscript! – pages 13-15, lines 472, 473, 475, 476, 485, 486, 498, 500, 502, 505): I think that the authors should use a common (one!) notation for figures citing, i.e. 5a (with no point between the number and the small case letter) – compare page 8 lines 307, 310, etc.

7.      Page 15 lines 506 and 509: It should be denoted “a region A (or B) in Fig. 7d – there is Fig 7.d.A.

8.      Page 18 lines 630-638: Please answer the questions: (i) How long did it take for the car to cross the sand ladder platform? (ii) Would not the sand ladder platform be damaged if the car had to stop on it for a few minutes? (iii) What is the maximum weight of the car/vehicle that the sand ladder platform can withstand?

9.      The minor errors and misprints found: (i) page 2 line 78: the word “European” should be deleted (two companies are out of Europe); (ii) page 5 line 178: there is “mp” and “p” should be the subscript; (iii) page 6 line 219: degree sign is underlined; (iv) page 7 the last line in Table 2: the length value is doubled; (v) page 8 lines 307: Figures should be cited consequently, i.e., Fig. 3, then Fig. 4, then Fig. 5, etc. The authors cited Fig. 3 and then Fig. 5; (vi) page 10 Table 5: the authors can add the error values for the calculated “X_%”; (vii) page 15 line 530: there should be (table 7 and Fig. 8c); (viii) page 17 Fig. 9 caption: there are two figures “c”, and there should be (d) – instead of the second “Fig. c”, and (e) instead of “Fig. d”.

Author Response

3. Point-by-point response to Comments and Suggestions for Authors

Comments 1: The manuscript entitled “Mechanical strength and surface analysis of a composite made from recycled carbon fibre obtained by the pyrolysis process for the development of a “Sand Ladder Platform” product” presents analyses of the recycled and commercial carbon fibres combined with the same resin/hardener system. The obtained data were compared. The authors conduct many experiments, i.e., the flexure, tensile, and Mode I ones. The manuscript is well-prepared, step-by-step, leading a potential reader throughout the measurements, preparation process, and testing process. I think the manuscript can be published in the Journal after minor amendments.

1.      Page 5, Eqs. (1)-(3): Why did the authors use the cross sign to denote multiplication? Please compare with Eq. (6) – there is a point sign, as it should be. The cross sign is used for vector multiplication (then one gets a vector quantity!).

Response 1: The authors appreciate the proposed suggestion. The changes can be found on page 3, highlighted in red.

Comments 2: Page 6 line 229 (and many others like this!): There is no “CF” in Eq. (4). There is “F_c” (“c” as the subscript) instead of.

Response 2: The authors appreciate the proposed suggestion. The changes can be found on page 6, line 229, highlighted in red.

Comments 3: Page 6 lines 245-252: Why did the authors choose this type of car (EcoSport, Ford)? Is this surreptitious advertising? The authors can use any other car, i.e. (more or less), about the same length, the length between the axis, and the total weight.

Response 3: The authors are grateful for the reviewer's comments. The car type specification is not an advertisement. I used my car to do the tests. I thought that if I put the car's specification in the paper it would be more faithful to what happened during the tests. According to the calculations for sizing the sand ladder platform, the product would support a weight force of 4110.39N (Table 9), which would be equivalent to approximately 420kg. The car used weighs approximately 433kg (Table 2).

Comments 4: Page 9 Table 3: Typically, up to two significant digits are given for the error value.

Response 4: The authors appreciate the proposed suggestion. The changes can be found on page 9, in Tables 3 and 4, highlighted in red.

Comments 5: Page 10 Eq. (10): see point 1!

Response 5: The authors appreciate the proposed suggestion. The changes can be found on page 10, highlighted in red.

Comments 6: Page 11 lines 380-381 (and many other in the manuscript! – pages 13-15, lines 472, 473, 475, 476, 485, 486, 498, 500, 502, 505): I think that the authors should use a common (one!) notation for figures citing, i.e. 5a (with no point between the number and the small case letter) – compare page 8 lines 307, 310, etc.

Response 6: The authors appreciate the suggestion. The changes can be found throughout the text, highlighted in red.

Comments 7: Page 15 lines 506 and 509: It should be denoted “a region A (or B) in Fig. 7d – there is Fig 7.d.A.

Response 7: The authors appreciate the proposed suggestion. The changes can be found throughout the text, highlighted in red.

Comments 8: Page 18 lines 630-638: Please answer the questions: (i) How long did it take for the car to cross the sand ladder platform? (ii) Would not the sand ladder platform be damaged if the car had to stop on it for a few minutes? (iii) What is the maximum weight of the car/vehicle that the sand ladder platform can withstand?

Response 8: The authors appreciate the proposed suggestion. The changes can be found on pages 18 and 19, highlighted in red.

Figure 11 shows images taken at the moment the Ecosport Freestyle 1.6 passes over the sand ladder platform and remains on the platform surface for 10 minutes. According to the Sand Ladder Platform calculations, the product would support a load of 4110.39N (Table 9), which is approximately 420kg. The car used weighs approximately 433kg (Table 2), so it can be concluded that the product is perfectly designed. A visual inspection was performed on the prototype after the test and no apparent damage was observed on the product.

Comments 9: The minor errors and misprints found: (i) page 2 line 78: the word “European” should be deleted (two companies are out of Europe); (ii) page 5 line 178: there is “mp” and “p” should be the subscript; (iii) page 6 line 219: degree sign is underlined; (iv) page 7 the last line in Table 2: the length value is doubled; (v) page 8 lines 307: Figures should be cited consequently, i.e., Fig. 3, then Fig. 4, then Fig. 5, etc. The authors cited Fig. 3 and then Fig. 5; (vi) page 10 Table 5: the authors can add the error values for the calculated “X_%”; (vii) page 15 line 530: there should be (table 7 and Fig. 8c); (viii) page 17 Fig. 9 caption: there are two figures “c”, and there should be (d) – instead of the second “Fig. c”, and (e) instead of “Fig. d”.

Response 9: The authors appreciate the proposed suggestion. The changes can be found throughout the text, highlighted in red.

4. Response to Comments on the Quality of English Language

Point 1: English language fine. No issues detected

Response 1:  NA

Round 2

Reviewer 2 Report

Comments and Suggestions for Authors

[The numbering of comments (C) is that used by the authors]

C1.  My argument was not about the demonstration of the success of carbon fiber recycling via formation of new composites but about the reference to a PARTICULAR application in the TITLE; the broad significance of the article rests on the fact that the fibers can be re-used for the fabrication of new composites with satisfactory mechanical properties, while the reference to the exact, and somewhat obscure sounding (: ‘sand ladder platform’), application of the new composite should be reserved for the text. I will not insist on that point because it has to do more with the way one wishes to ‘advertise’ his/her work and enhance readership, especially in view of the fact that this is a Materials journal and not, specifically, a Composites journal.

C4. My question is about the origin of drop of modulus; hence the fact that two related moduli drop similarly is not an answer – the issue is: ‘Why do both moduli drop?’.

Please, consider possible reasons such as contributions of porosity, weakening of the fiber-matrix interface (/undermining of the fiber-matrix adhesion), effective shortening of the fibers (because of cracks). The authors do mention voids and inferior adhesion (new lines 411-414) as potential reasons for inferior mechanical properties but this is a sweeping statement that does not offer the desirable level of insight as different mechanical properties are not affected 'similarly' by the same factors (as I already mentioned in my original comment) and some less generic references to value drop of [all] mechanical properties is expected.

As regards language, I suggest some careful reading and recasting of the new phrases, e.g.: (a) The sand ladder platform was chosen to be developed. (Line 24 does not blend well with ‘surrounding’ phrases) and (b) […] there has been an increase in the development […] (Lines 33-34; yet development does not increase).

Overall comment: The article if published as it stands it will be better (e.g. richer as regards data) than several somewhat related and already published articles, but this is a good reason to spend a little more time on it; a non-haphazard Minor Revision  can  make it somewhat more attractive.

Comments on the Quality of English Language

(Repetition of a preceding comment:)

As regards language, I suggest some careful reading and recasting of the new phrases, e.g.: (a) The sand ladder platform was chosen to be developed. (Line 24 does not blend well with ‘surrounding’ phrases) and (b) […] there has been an increase in the development […] (Lines 33-34; yet development does not increase).

Author Response

Dear Reviewer,

The authors would like to thank the Reviewer for taking the time to review this paper and provide positive feedback, useful suggestions, and valuable criticisms. We have carefully considered the reviewer’s comments and believe the paper has been improved.

Below you can find the Reviewer’s questions, as well as our responses/actions.

Comments 1: My argument was not about the demonstration of the success of carbon fiber recycling via formation of new composites but about the reference to a PARTICULAR application in the TITLE; the broad significance of the article rests on the fact that the fibers can be re-used for the fabrication of new composites with satisfactory mechanical properties, while the reference to the exact, and somewhat obscure sounding (: ‘sand ladder platform’), application of the new composite should be reserved for the text. I will not insist on that point because it has to do more with the way one wishes to ‘advertise’ his/her work and enhance readership, especially in view of the fact that this is a Materials journal and not, specifically, a Composites journal.

Response 1: The authors appreciate the proposed suggestion. The changes can be found on page 1, highlighted in red.

The title was changed to “Mechanical strength and surface analysis of a composite made from recycled carbon fibre obtained by the pyrolysis process for reuse in the manufacture of new composites”.

Comments 2: My question is about the origin of drop of modulus; hence the fact that two related moduli drop similarly is not an answer – the issue is: ‘Why do both moduli drop?’.

Please, consider possible reasons such as contributions of porosity, weakening of the fiber-matrix interface (/undermining of the fiber-matrix adhesion), effective shortening of the fibers (because of cracks). The authors do mention voids and inferior adhesion (new lines 411-414) as potential reasons for inferior mechanical properties but this is a sweeping statement that does not offer the desirable level of insight as different mechanical properties are not affected 'similarly' by the same factors (as I already mentioned in my original comment) and some less generic references to value drop of [all] mechanical properties is expected.

Response 2: The authors appreciate the proposed suggestion. The changes can be found on page 10, lines 380-391 and pages 12 and 13, lines 439-485, highlighted in red, respectively.

Besides that, the rCFRP had a reduction of around 17.3% in relation to the EB and a reduction of around 13.2% in relation to the maximum stress. The reduction in the mechanical properties was expected in rCFRP. It can therefore be guaranteed that composites made with rCRFP will have lower values for all their mechanical properties compared to composites made with commercial fibres. The decrease in EB may be related to the loss of sizing and the consequent decrease in fibre-adhesive matrix interaction, which would lead to a decrease in the interfacial shear strength of the matrix [39]. Furthermore, the chosen manufacturing method may have introduced defects such as fibre misalignment and the number of voids present in the specimens, as can be seen in Figure 5, which shows that the values found are realistic, but that the manufacturing method used needs to be optimised to obtain specimens with better mechanical properties.

The values for the maximum stress were around 36.02% and for the strain at the maximum stress, around 30.22% were relatively higher than the values presented in the flexural test, demonstrating that for purely tensile loads the rCFRP are significantly affected. However, the reduction of about 17.75% in relation to the modulus of elasticity was similar to the reduction shown in the flexural test, this means that the fiber capacity to deform elastically remained the same, while deformation and rupture are proportionally smaller, when the values obtained for the specimens manufactured with commercial carbon fibre were compared with the technical data values for commercial prepreg. Anyway, it must be taken into account that the resin contained in the prepreg (Epoxy Resin 8552) was different from the resin system used in the manufacture of the Araldite® LY 5052/AradurTM 5052 test specimens, this being one of the factors that contributes to the difference of values. Table 4 shows the technical data of the prepreg provided by the manufacturer.

Alves et al. [45] subjected the recycled composite by pyrolysis to tensile tests and observed the same behaviour reported here They observed that the tensile modulus was very close for the two laminates. On the other hand, the tensile strength and ultimate longitudinal strain are about 40% lower in the recycled specimens. The loss of tensile strength at this point may be attributed to a combination of the damage on the surface of the fiber shown on micrography and the voids on the resin-rich areas. Most of the tensile load carried by a carbon fiber is transmitted in the fiber surface; any disruption to the surface can result in a mass-disproportionate loss of tensile properties. It was concluded that this reduction in shear stress and tensile strength is due to the weak interaction between the matrix and the resin because of the sizing loss.

Feraboli et al. [38] studied the mechanical properties of the recycled composite obtained by a chemical process and they observed a decrease in tensile strength flexural approximately 25– 30% and flexure strength approximately 70% and 52% when compared with results obtained for those manufactured with commercial carbon fibre.

According to Pimenta and Pinho [6, 7] pyrolysis process is a good recycling process because it has a high retention of mechanical properties, the potential to recover chemical feedstock from the resin, and no use of chemical solvents. They also observed a decrease in mechanical performance by approximately 20% in the elastic properties of the recycled composite.

In addition to these factors, another issue that can affect the reduction in modulus values is the alignment of the fibres in the final composite. As mentioned above, the pyrolysis process of carbon fibres removes the matrix and sizing, both of which are important in maintaining fibre alignment. After the pyrolysis process, the carbon fibres removed from the oven have no surface protection and can cause breakage and misalignment. This misalignment can be exacerbated by the manufacturing process chosen, in this case, the hand layup method. This method consists of impregnating the fibre surface with resin using a spatula or impregnation roller. The back-and-forth movement of the object can cause some misalignment of the fibres. This factor drastically affects the tensile modulus results; van de Werken et al. [46] found that samples with sizing but no alignment had a 35% lower modulus than those without sizing but with alignment. In the model described by van de Werken et al. [46], the composite modulus increases monotonically as the fibres become more aligned along the tensile axis. Turner et al [47] demonstrated in their studies that fibre alignment is a critical factor in attaining high mechanical properties and high recovered fibre utilization.

The references used to develop the discussion of modulus of elasticity and Young's modulus were:

·          Jiang, G., Pickering, S. J. Structure–property relationship of recycled carbon fibres revealed by pyrolysis recycling process. Journal of Materials Science, 2015, 51(4), 1949–1958.

·          van de Werken N., Reese M.S., Taha M. R., Tehrani M., Investigating the effects of fiber surface treatment and alignment on mechanical properties of recycled carbon fiber composites, Compos. Part A: Appl. Sci. and Manuf., 2019, 119, 38-47.

·          Turner TA, Warrior NA, Pickering SJ. Development of high value moulding compounds from recycled carbon fibres. Plast Rubber Compos 2010;39(3–5), 151-156.

Overall comment: The article if published as it stands it will be better (e.g. richer as regards data) than several somewhat related and already published articles, but this is a good reason to spend a little more time on it; a non-haphazard Minor Revision can make it somewhat more attractive.

Response: The authors would like to thank once again all the valuable contributions given by the Reviewers, allowing for the paper’s improvement.

Response to Comments on the Quality of English Language

Point 1: As regards language, I suggest some careful reading and recasting of the new phrases, e.g.: (a) The sand ladder platform was chosen to be developed. (Line 24 does not blend well with ‘surrounding’ phrases) and (b) […] there has been an increase in the development […] (Lines 33-34; yet development does not increase)

Response 1:  The authors appreciate the proposed suggestion. The changes can be found on page 1, lines 22-23 and lines 32-34, highlighted in red.

The sand ladder platform was the project chosen for the development of a product made with recycled carbon fibre.

In some industries, such as aerospace, automotive and sports, there has been an increase in the use of composite materials to produce more efficient structures and more complex designs.

We are looking forward to hearing from you.

Thank you so much for your attention.

Kind regards,

Rita Sales
